# Determinants of HIV Pre-Exposure Prophylaxis (PrEP) Retention among Transgender Women: A Sequential, Explanatory Mixed Methods Study

**DOI:** 10.3390/ijerph21020133

**Published:** 2024-01-25

**Authors:** Jack Andrzejewski, Heather A. Pines, Sheldon Morris, Leah Burke, Robert Bolan, Jae Sevelius, David J. Moore, Jill Blumenthal

**Affiliations:** 1San Diego Joint Doctoral Program in Public Health, San Diego State University—University of California San Diego, San Diego, CA 92093, USA; 2School of Public Health, San Diego State University, San Diego, CA 92182, USA; hpines@sdsu.edu; 3Herbert Wertheim School of Public Health and Human Longevity Science, University of California San Diego, La Jolla, CA 92093, USA; 4Department of Medicine, University of California San Diego, La Jolla, CA 92161, USA; shmorris@health.ucsd.edu (S.M.); lbburke@health.ucsd.edu (L.B.); jblumenthal@health.ucsd.edu (J.B.); 5LA LGBT Center, Los Angeles, CA 90038, USA; bbolan@lalgbtcenter.org; 6Department of Psychiatry, Columbia University, New York, NY 10032, USA; j.sevelius@ucsf.edu; 7Department of Psychiatry, University of California San Diego, La Jolla, CA 92093, USA; djmoore@health.ucsd.edu

**Keywords:** PrEP, transgender women, HIV, sex work, substance use, gender affirmation

## Abstract

Transgender women (TW) face inequities in HIV and unique barriers to PrEP, an effective biomedical intervention to prevent HIV acquisition. To improve PrEP retention among TW, we examined factors related to retention using a two-phase, sequential explanatory mixed methods approach. In Phase I, we used data from a trial of 170 TW who were provided oral PrEP to examine predictors of 24-week retention. In Phase II, we conducted 15 in-depth interviews with PrEP-experienced TW and used thematic analysis to explain Phase I findings. In Phase I, more participants who were not retained at 24 weeks reported sex work engagement (18% versus 7%) and substantial/severe drug use (18% versus 8%). In Phase II, participants reported drug use as a barrier to PrEP, often in the context of sex work, and we identified two subcategories of sex work. TW engaged in “non-survival sex work” had little difficulty staying on PrEP, while those engaged in “survival sex work” struggled to stay on PrEP. In Phase I, fewer participants not retained at 24 weeks reported gender-affirming hormone therapy (GAHT) use (56% versus 71%). In Phase II, participants prioritized medical gender affirmation services over PrEP but also described the bidirectional benefits of accessing GAHT and PrEP. TW who engaged in “survival sex work” experience barriers to PrEP retention (e.g., unstable housing, drug use) and may require additional support to stay in PrEP care.

## 1. Introduction

From 2019 to 2020, the Centers for Disease Control and Prevention estimated the prevalence of HIV among transgender women (TW) to be 42% [1,2], indicating a need to better understand the uptake and retention of HIV prevention strategies among this population. Pre-exposure Prophylaxis (PrEP) for HIV is an effective biomedical intervention that can reduce the risk of HIV infection when taken as prescribed [3]. PrEP is recommended for use by people at increased risk of acquiring HIV, including sexually active transgender persons [1]. While about 92% of HIV-negative TW were aware of PrEP, only about 32% currently used PrEP [1]. Further, retention in PrEP care remains suboptimal, indicating a need for more research on factors associated with PrEP engagement and retention, which can inform prevention efforts [4].

Health inequities among TW can in part be attributed to cisgenderism, the pervasive system that privileges cisgender people over transgender people [5]. Transgender people face unique barriers to HIV prevention services, such as PrEP, including structural factors that necessitate participation in survival sex work and limit access to gender affirmation services, including gender-affirming health care [6]. Further, problematic substance use behaviors that may result from a lack of access to gender affirmation can disrupt engagement in PrEP [7,8]. Yet limited research describing how or why these factors contribute to PrEP disengagement exists, warranting additional investigation to increase PrEP uptake and adherence [2,9,10].

TW face disparities in access to social determinants of health including employment, stable housing, and health insurance, all of which may contribute to increased participation in sex work and decrease PrEP engagement [10,11,12,13,14]. Among TW in the US, prevalence estimates were 32% for unstable housing, 39% for employment, and 72% for health insurance [15]. TW also report experiencing discrimination in health care and a lack of access to culturally competent care [16]. Lack of access to social determinants of health may lead to sex work as a means of survival [17,18]. Participation in sex work may increase the risk of acquiring HIV [19], particularly when the agency to use condoms is diminished due to power imbalances between sex workers and their clients who may prefer not to use condoms [20,21]. Thus, sex work may be a motivating factor for PrEP among TW. Although some TW engaged in sex work report receiving peer support for PrEP use [22], more research is needed to understand the structural drivers of PrEP engagement among this population.

Gender affirmation is a transgender-specific social determinant of health and has been defined as the social process of being recognized and supported in one’s gender identity, expression, and/or role [23]. Gender affirmation includes at least four distinct constructs, which are social, psychological, medical, and legal affirmation [23]. The Gender Affirmation Framework describes how a need for and lack of access to gender affirmation can lead to high-risk social contexts and behaviors, including survival sex work, substance use, and lack of engagement in medical care [7]. Medical gender affirmation, such as the use of gender-affirming hormone therapies (GAHTs) and surgical procedures, has been associated with PrEP awareness and PrEP use [24,25,26]. Qualitative research has suggested that lack of access to gender affirmation is an important determinant of PrEP care and that co-location of services may help facilitate PrEP engagement [27,28]. Further, concerns about drug–drug interactions between GAHT and PrEP have been noted by TW [29]. Yet, research suggests that PrEP does not have an impact on GAHT, and although GAHT may reduce PrEP drug concentrations, this effect is small in magnitude, and drug levels remain within the therapeutic range [30]. However, more work is needed to understand how and why gender affirmation is related to PrEP use among TW.

Our study begins to fill these gaps in the literature by examining engagement in PrEP care among TW in relation to sex work and gender affirmation using an explanatory, sequential mixed methods approach. First, we aimed to examine factors related to engagement in PrEP care among TW not living with HIV who participated in a PrEP-focused randomized controlled trial. Second, we aimed to explore how and why sex work and gender affirmation were related to PrEP engagement through interviews with PrEP-experienced TW not living with HIV.

## 2. Materials and Methods

### 2.1. Study Design

Consistent with our research aims, we used a sequential, explanatory mixed methods approach [31], which included two phases: (I) analysis of quantitative data from the iTAB Plus Motivational Interviewing for PrEP (iMPrEPT) study [32] and (II) qualitative in-depth interviews. In Phase I, we analyzed quantitative data collected from the 170 TW (i.e., assigned male sex at birth and identifying as TW or non-binary persons) enrolled in iMPrEPT to examine factors related to continued PrEP care engagement. iMPrEPT was a 48-week randomized controlled trial evaluating brief motivational interviewing along with a text message-based adherence intervention among 255 transgender individuals in San Diego and Los Angeles who were provided oral PrEP with emtricitabine/tenofovir disoproxil fumarate (FTC/TDF) throughout the study starting at baseline from June 2017 to September 2020 [32]. Participants assigned female at birth were excluded from the current analysis. To be eligible for iMPrEPT, participants were transgender or nonbinary, 18 years or older, HIV-negative, and had one of the following risk factors for HIV: (1) 1 or more HIV-positive sexual partners in the past 4 weeks or more than 1 HIV-positive sexual partner in the past year; (2) anticipated condomless sex with a sexual partner assigned male at birth in the next 3 months; or (3) any sexual partners assigned male at birth in the past year and at least one of the following in the past year: (1) condomless sex; (2) any sexually transmitted infections (STIs); (3) participation in exchange sex; or (4) postexposure prophylaxis use. Participants were excluded if they exhibited active hepatitis B infection. Participants were recruited through (1) a Trans Community Advisory board; (2) community events including a trans health and wellness day, National Transgender HIV Testing Day, and a safe space clothing swap; (3) online via a study-specific website and social media accounts; (4) engagement with local trans-focused community organizations; (5) referrals from providers; and (6) word of mouth.

Phase I findings were used to inform the recruitment of TW with sex work or transactional sex experience to participate in Phase II qualitative interviews to contextualize Phase I findings and elicit other potential factors related to PrEP engagement. Findings from Phase I also informed the development of the Phase II in-depth interview guide. Fifteen PrEP-experienced TW were purposively sampled with about half currently on PrEP. Recruitment was conducted via social media in collaboration with a local trans community organization. Participants in Phase II were not recruited from Phase I participants, thus it is unlikely that Phase II participants were in Phase I of the study. Although participants in Phase II were likely not in Phase I, they were recruited from the same target population. This is referred to as a parallel, rather than identical or nested, sampling design, which is appropriate for sequential mixed methods studies [33], which we chose for feasibility.

### 2.2. Data Collection

For Phase I, quantitative data were collected at baseline and 24-week follow-up visits as part of the iMPrEPT study as previously described [32]. Follow-up visits occurred every 12 weeks to align with recommendations for PrEP refill visits up to 48 weeks. Baseline surveys were administered via computer-assisted self-interviewing (CASI) in English or Spanish depending on the participant’s language preference and collected information on socio-demographics (age in years, race/ethnicity, sexual orientation, relationship status, highest level of education completed, employment status, and health insurance status), engagement in sex work as full- or part-time employment, substantial or severe drug use (defined as a score ≥6 on the Drug Abuse Screening Test) [34], and GAHT use. Retention in the iMPrEPT study at 24 weeks was used as a proxy measure of continued PrEP care engagement.

For Phase II, we conducted in-depth interviews using a semi-structured interview guide, which covered the topics of gender affirmation, sex work/transactional sex, PrEP experiences, barriers and facilitators of PrEP engagement, and the relationship between gender affirmation and PrEP care. Interviews also elicited ways participants have overcome barriers to PrEP continuation and additional supports that might help them overcome barriers to PrEP reengagement, re-initiation, and retention. The interview guide included open-ended questions, for example, “What have been your experiences with transactional sex?”, “What, if any, challenges have you faced with taking PrEP?”, and “If it were easier to access gender affirmation services, how if at all would this help you in terms of PrEP?” Interviews were audio-recorded, transcribed verbatim, and iteratively reviewed to inform additional probing in subsequent interviews.

### 2.3. Analysis

For Phase I, we calculated descriptive statistics to characterize iMPrEPT participants with respect to 24-week retention in the iMPrEPT study overall and by socio-demographics, engagement in sex work, substantial or severe drug use, and GAHT use.

For Phase II, we applied thematic analysis based on strategies outlined by Guest, MacQueen, and Namey (2012) [35]. Transcripts were uploaded into Dedoose Version 9.0.17. (SocioCultural Research Consultants, LLC, Los Angeles, CA, USA) qualitative data analysis software for coding [36]. First, we developed a codebook based on the study objectives of understanding barriers and facilitators to PrEP care. We also included codes on the relationship between transactional sex and PrEP care and the relationship between gender affirmation and PrEP care. Three coders used an iterative process of coding and discussion on three transcripts to refine the codebook until a consensus on the coding application was achieved. Two coders then independently coded each transcript. Discrepancies reflected missing a code from one of the two coders and not a disagreement of coding application and, thus, were resolved by the lead author.

After coding, we examined coded excerpts to identify themes. We used analytic memos to develop themes by assessing repetition and linguistic connectors (i.e., because, if, since), and through constant comparison of coded segments. Themes were then refined through open discussion among the study team. During thematic analysis, we reflected on the findings from Phase I and integrated these findings into the presentation of the themes from Phase II.

Priority for data analysis and interpretation was given to Phase II because Phase I was a secondary analysis, and Phase II data were collected by the study team for the purpose of understanding barriers and facilitators of PrEP engagement, which also informed the development of a pilot intervention currently underway to reengage, re-initiate, and retain TW in PrEP care. Further, qualitative data collection focused on explaining PrEP engagement in relation to factors identified from Phase I.

## 3. Results

Sociodemographic information for Phase I participants is provided in Table 1. At baseline, the mean age of Phase I participants was 33.1 years (SD 10.2), with 35.7% Latinx, 20.8% White, and 14.3% Black individuals. Eleven percent of participants reported engagement in sex work, eleven percent reported severe or substantial substance use problems, and sixty-six percent reported using GAHT. Sociodemographic information for Phase II participants is reported in Table 2. The mean age of Phase II participants was 35.8 years old (SD 9.7), with 40% Latinx, 60% White, and 20% Black individuals. All participants had PrEP experience with 67% currently on PrEP, and 60% had engaged in sex work or transactional sex.

### 3.1. Sex Work


*In Phase I, a greater proportion of participants who were not retained at 24 weeks reported engaging in sex work than those who were retained (18% vs. 7%) (Table 1). Phase II provided context as to why some TW engaged in sex work had difficulty staying engaged in PrEP care. In Phase II, two subcategories of sex work engagement were identified. The first was characterized as engaging in “Non-Survival Sex Work”—these TW had little difficulty staying in PrEP care, sought clients from online sources, had stable housing, accessed GAHT through providers, and exchanged sex primarily for money. The second was characterized as engaging in “Survival Sex Work”—these TW struggled to stay in PrEP care, had street-based clients, were unstably housed, used black market hormones, and more frequently exchanged sex for drugs.*


Sex work was a common motivator for PrEP use, as one participant described: “if I was gonna be engaging in transactional sex, like it, it… it was just a no brainer to… to be taking that [PrEP]” (32 years old, on PrEP, engaged in sex work). This sentiment seemed at odds with the retention seen in Phase I. Further analysis of Phase II data revealed that some TW who engaged in sex work in Phase II had little trouble staying engaged in PrEP care (i.e., those who engaged in non-survival sex work) while others faced significant barriers to PrEP (i.e., those who engaged in survival sex work).

#### 3.1.1. Non-Survival Sex Work

One participant described how sex work was something she could fall back on if she faced discrimination in other forms of employment or if she needed extra money to cover bills. Even though she had an Associate’s degree in vocational nursing, she continued to do sex work as needed. She was open with her current doctor (who prescribes her GAHT) for the first time about participating in sex work, and when a condom broke with a client, her provider initiated PrEP. Although she previously did not use condoms at times with clients, she now always uses condoms even though she is on PrEP. Condoms were also important to her to protect her partner from STIs and HIV, as she explained:
“Now I always. Uhmm. I always use condoms for everything. You know. Usually, I would be like if I knew the client or whatever maybe I would do oral without it. But now I do not even do that anymore. I just it’s automatic. You know what I mean? Cause it’s just not me, I have to think about him. You know and that really scared me. You know. When she [the doctor] said I had gonorrhea and chlamydia, I was fucking really scared. You know. And then, I was worried about him [her partner]”(47 years old, on PrEP, engaged in sex work).
Besides initially experiencing headaches when taking PrEP, she did not experience other challenges: “Nothing, I’ve nothing. Oh! Ahm. No, not even the financial thing, no. Cause ahm the insurance covered it. I only pay like 4 dollars or something” (47 years old, on PrEP, engaged in sex work).

Another participant only performed sex work for money and has never needed to for shelter or food, because she had friend and family support: “I only do it for the money. For shelter, food I don’t need to do that. Cause I know I have family that will just or friends who will like, if I need food, they’ll help me” (26 years old, on PrEP, engaged in sex work). To protect her health, she uses PrEP and condoms consistently. When asked what challenges she faced with PrEP, she responded, “Nothing”.

#### 3.1.2. Survival Sex Work

On the other hand, some transgender women reported engaging in sex work because they do not have other employment options due to discrimination. One participant stated:
“There has been so many times where I’ve applied for jobs, and they told me that I couldn’t work there once they’d seen my ID. Uhm, sex work I consider is a form of survival… There’s been plenty of times where I’ve had to have sex with people just to pay my rent at the end of the month or even just to stay the night at somebody’s place just to survive”(26 years old, on PrEP, engaged in sex work).
This participant later described how cost was a major barrier to PrEP, which leads TW to make trade-offs between protecting themselves from HIV and achieving their gender affirmation goals, which could lead to more income through sex work as clients sometimes prefer TW who have undergone more gender-affirming medical treatments. She also noted that TW often access hormones and silicone injections outside of medical settings to save money.

Another participant described being homeless due to an unsafe living situation related to her gender and sexuality, which made it difficult to maintain other forms of employment leading to engaging in sex work. While she remarked there was no shame in doing sex work, trans folks often have no other option: “I wouldn’t not do it again, but most of the time it is just because of the underlying conditions in society that make it difficult to have a conventional type of employment” (28 years old, on PrEP, engaged in sex work). This participant reported trouble with adherence due to not having PrEP with her and forgetting to take it.

One participant talked about engaging in sex work to get access to hormones through non-medical sources, which was difficult during the COVID-19 pandemic. She had not been able to stay on PrEP consistently and although she asked people to use condoms, they did not always want to, and she was not able to say no because they had something she needed. When asked how she prioritizes her sexual health in the context of sex work, she stated:
“Not well. I am. I’ve asked for them to wear condoms. I’ve asked of them to be respectful in that manner, but if they uhm… refuse I don’t really know how to say no. Because, I should have a hard, a hard line that I don’t want them to cross that, because I don’t want them to cross that line. But, when they do, I don’t know how to just say ‘No, this isn’t happening’ like because at the end of the day they have something I want, or need, and… It’s… I’m playing by their rules”(28 years old, not on PrEP, engaged in sex work).
Further, she reported being unable to attend PrEP appointments, which led to her discontinuing PrEP use, and she has yet to re-initiate PrEP despite wanting to.

Other participants reported exchanging sex for food or drugs: “I’ve done sort of off and on, here and there. Uh, never like consistently or or routinely. But, just sort of like one off or opportunistically” (35 years old, not on PrEP, engaged in sex work). After losing her job and insurance, she was no longer able to access PrEP.

Regardless of the type of sex work in which TW were engaged, housing resources were noted to be a vital need for this population. One participant described:

“They’re scared to go fill out an application for a house cause they don’t have a W2. Or their money is coming from sex work. They can’t, you know. It’s just. I, I just think that people need to, you know, just be open minded about, you know. Transexuals are out there. You know, they they are escorting and they do make money they can pay their bills. You know, so maybe help them get into housing and stuff like that, cause, it’s a lot. That’s what happened to me a lot. I couldn’t, you know, go and fill an application. They’re gonna be like ‘Okay, well where is your pay stubs? Where is your W2s?’ And, you know. I could be making more money than a lot of people but I can’t prove it. You know, so. And housing is a big thing for transexuals. A lot of them don’t have it”(47 years old, on PrEP, engaged in sex work).

Even when TW have enough income to pay rent, they may not be able to obtain a lease due to their income coming from sex work. Privacy and discretion were also mentioned as important factors for sex workers to prevent violence.
“I mean, as far as like sex work goes it’s uh, it’s, it’s something that people often have to like hand like they have to go about it with a level of privacy and uh discretion. So, just like. Is like as long as the other people can have that discretion as well. And not, not in like a shameful way but just in like you have to be strategic and uhm people, people can be hostile towards you. So, just. It doesn’t have to be like screamed at you but uh just being able, so being able to relate that relate stuff to you in a way that uhm is secure”(28 years old, on PrEP, engaged in sex work).
This quote suggests that clinical services and health promotion interventions targeting TW engaged in sex work should consider the risk of exposing the patient’s/participant’s sex work status, as this may lead to violence against them.

### 3.2. Substance Use


*In Phase I, a greater proportion of participants who were not retained at 24 weeks reported substantial or severe substance use than those who were retained (18% vs. 8%) (Table 1). Phase II provided insights into the relationship between PrEP and substance use. In Phase II, participants reported substance use as a barrier to PrEP, often in the context of sex work.*


Some participants described engaging in sex work primarily to obtain substances. “I should just say it was like, it was drug related. Like I was exchanging like sex for for like drugs” (35 years old, not on PrEP, engaged in sex work). Substance use was seen as negatively impacting safer sex practices such as condom use: “Uh, but to have it’s been times and it’s in part due to my inhibition being almost nothing due to drug use. Uh I’ve had unprotected sex. So, not made my health priority in most cases at all” (35 years old, not on PrEP, engaged in sex work).

Others described how many of their sex work clients used substances, even when they did not, and one participant stated other sex workers she knew did use substances during sex work. In one case, a participant would bring sterile syringes with her to help facilitate harm reduction among her clients, even though she did not use substances herself. “I never did drugs like that. But I knew that there was people that I was sleeping with that were doing drugs that I just didn’t feel comfortable with them sharing needles. So, I always had needles on the side to give them. So that they would have clean needles whenever they used the drugs” (26 years old, on PrEP, engaged in sex work).

One participant expressed they are more comfortable talking with people who have themselves engaged in substance use behaviors: “It’s fine to just sort of have candid conversations about transactional sex or drug use with other whom I already know are engaged in such behavior… In addition to uh expressing empathy and love uh to sort of understanding the circumstances that would lead someone to display in that behavior” (35 years old, not on PrEP, engaged in sex work). Even when providers have not engaged in such behaviors, this suggests that it is important for gender-affirming providers to not stigmatize substance use and be able to express empathy for those using substances. 

### 3.3. Gender Affirmation


*In Phase I, a smaller proportion of participants not retained at 24 weeks reported GAHT than those who were retained (56% vs. 71%) (Table 1). Phase II described how PrEP and GAHT were related for TW. In Phase II, participants consistently prioritized GAHT over PrEP, describing GAHT as a basic necessity and lifesaving, yet many also described the bidirectional benefits of accessing GAHT and PrEP.*


Offering PrEP, GAHT, and other medical gender affirmation services together was consistently cited as a potential strategy to improve PrEP care. Although participants had trouble at times attending PrEP appointments or obtaining PrEP prescriptions, they generally reported less trouble with obtaining GAHT. One participant shared, “I’m terrible at obtaining my prescriptions. Terrible. It doesn’t matter what prescription. Aside from HRT, I’m actually pretty good with that one. But, anything else I’m terrible at” (28 years old, not on PrEP, engaged in sex work). Taking PrEP alongside GAHT also facilitated adherence to the medication. “Because I took my hormones every day and I got anxiety when I didn’t only because I’ve felt more masculine whenever I did and that was something that I wasn’t comfortable with. So, always placing my PrEP there was the thing” (26 years old, on PrEP, engaged in sex work). In this case, the gender-affirming benefits of GAHT motivated medication adherence.

When asked directly which they would prioritize between PrEP and gender affirmation services, participants nearly unanimously chose gender affirmation services. One participant reported, “I just care about, feeling complete, you know. I just, I just want my tran- I just I just. Being who I am is the most important thing in my life. And, just, ah, fulfilling my transition and uhm…That’s re- that’s the most important thing to me and then t- My health should be important. But just being who I am first, makes me healthy” (47 years old, on PrEP, engaged in sex work). Only one participant prioritized PrEP over gender affirmation services, possibly because she did not require medical gender affirmation services, instead emphasizing her need for psychological and social gender affirmation. One participant could not prioritize one over the other, because she felt both were important to protect her health. Since medical gender affirmation services are often not fully covered by insurance, the financial cost of PrEP may represent a tradeoff between gender affirmation and sexual health-related goals. Getting paid financial incentives for PrEP could also help with gender affirmation, for example: “I mean I would use it [financial incentives for PrEP] to like buy clothes for myself which would be, I mean, gender affirming in a way” (35 years old, not on PrEP, not engaged in sex work).

Gender affirmation was usually the primary goal, as it impacted both mental and physical health: “If you don’t stay on your hormones and you don’t stay on top of it, it affects you. Like emotionally and physically, like you know like… that’s why it’s like I would if I had to choose, I choose my hormones before my PrEP” (26 years old, on PrEP, engaged in sex work). Thus, improving access to gender affirmation services could allow participants to focus on other aspects of their health such as HIV prevention. Conversely, increased access to PrEP could help by allowing TW the ability to focus on gender affirmation: “it’s [PrEP] given me like an ease of mind. Uhm, like just a little bit of peace too. It’s uh part, I mean cause it’s I mean gender is is, it takes a toll on you and like the having to work through dysphoria and stuff is is uhm a mentally strenuous process. So, not having to think about ‘I’m gonna die of AIDS’ has definitely [laugh] made it easier to deal with dysphoria and stuff” (28 years old, on PrEP, engaged in sex work). While some participants felt that feeling feminine was more important than having sex, others felt that sex with men and taking on a “more traditional feminine role” (62 years old, not on PrEP, not engaged in sex work) during sex was an important aspect of their gender affirmation, with PrEP as a key facilitator. One participant shared, “I meet a whole, like array of, of different men and… each one is different and, and… uhm… I guess… I guess the fact that I… that so many different good-looking men wanna come over and meet me is, is… you know it’s gender affirming, I guess…Uhm, and then, uh, I wouldn’t be able to like, I wouldn’t feel comfortable meeting all these people if I didn’t have PrEP” (32 years old, on PrEP, engaged in sex work). For this participant, PrEP had a positive impact on her gender affirmation.

## 4. Discussion

Our findings suggest that although TW engaged in sex work may have difficulty engaging in PrEP care, this difficulty may be related to extraneous factors including lack of access to food, housing, employment, social support, and gender affirmation services, and not inherently related to sex work. In contrast, TW who engaged in sex work in our study and who had access to food, housing, employment, social support, and medical gender affirmation services reported little or no trouble engaging in PrEP care. Previous research has found that structural factors, such as HIV stigma, history of incarceration, access to gender-affirming care, health insurance, housing instability, and systemic racism are barriers to PrEP use among TW engaged in sex work [6,14,37]. Our findings extend previous research by suggesting that a lack of access to social determinants of health is a driver of PrEP disengagement among TW engaged in survival sex work, but less so for those engaged in non-survival sex work. Further, the criminalization of sex work may also increase vulnerability to HIV [27], potentially exacerbating the impact of barriers to PrEP engagement among TW engaged in sex work. Sex work is work, and the criminalization of sex work does not reduce it [38].

Our findings support previous research citing substance use as a barrier to PrEP engagement [8,14,26,39], particularly for those with substantial or severe substance use problems. Further, substance use is associated with greater HIV risk behaviors among transgender women, indicating a need for PrEP [40]. Participants also described how the context of sex work can lead to substance use, as clients may be using drugs and encouraging TW engaged in sex work to partake, particularly during survival sex work. Yet some TW may abstain from such behaviors and even help promote harm reduction practices, such as providing sterile syringes to their clients. Provider stigma was also noted as a barrier to discussing substance use. Given the potential for substance use to negatively impact PrEP engagement, healthcare providers may benefit from interventions to reduce stigma against substance use [41]. TW who use substances may also benefit from the colocation of harm reduction services with PrEP care [42].

Gender affirmation remains vitally important to the provision of PrEP services [29]. Consistent with previous findings [37], our study found that TW consistently prioritize gender affirmation over PrEP, as gender affirmation was considered by some to be “a basic necessity” and “lifesaving”, affecting both physical and mental health. Additionally, PrEP was described as a facilitator of gender affirmation, as having sex with men may be an important aspect of gender affirmation for some TW, with PrEP helping them to feel more comfortable and protected during sex. Not having to worry about HIV helped TW focus on their gender affirmation-related goals and similarly, not having to worry about gender affirmation-related goals could allow TW to focus on engaging in PrEP care. Thus, our study identified potential impacts of gender affirmation on PrEP and vice versa, suggesting a bidirectional relationship.

Given the potential costs of medical gender affirmation services and PrEP, particularly for those who are uninsured, TW facing economic precarity may be forced to consider tradeoffs between the two, likely choosing medical gender affirmation services. Thus, interventions that offset the financial costs of PrEP or medical gender affirmation services may be mutually beneficial. Finally, our findings emphasize the importance of colocation of PrEP care and medical gender affirmation services, in alignment with what others have suggested [28,43,44]. Colocation can support PrEP adherence because TW report being better at attending medical appointments related to gender affirmation services than PrEP, likely due to their motivation for achieving gender affirmation-related goals. Colocation of services can also reduce the number of medical appointments TW need to attend. In addition, providers trained in offering gender affirmation services may be more affirming in the rest of their medical practice including sexual health care.

Future research investigating the relationship between social determinants of health and PrEP engagement among TW who participate in sex work is needed. Interventions should consider the colocation of other supportive services with PrEP care, including services that address gender affirmation, harm reduction, food security, housing, employment, and social support.

### Limitations

Our findings may not generalize beyond our study cohort of transgender women living in Southern California, given that our study did not consider a probability sample of our target population. Further, participants in Phase II were not included in Phase I, thus the explanation of Phase I findings may not reflect the experiences of Phase I participants. However, participants in both phases were sampled from the same target population. In Phase II, we asked participants about substance use only in the context of sex work, which may have limited discussions of substance use in the context of sex work. However, some participants initiated discussions of substance use outside the context of sex work. Phase II data were collected during the COVID-19 pandemic, which may have impacted participants’ current experiences of sex work, substance use, and gender affirmation on PrEP engagement. However, few participants in Phase II discussed the COVID-19 lockdown and its impacts on PrEP, limiting our ability to interpret how it may impact our study findings more broadly. Given our findings’ alignment with research conducted before COVID-19, this may suggest that our results are robust to time outside the COVID-19 pandemic. 

## 5. Conclusions

Although TW who engage in sex work may face barriers to PrEP engagement, this finding may be due to a lack of access to social determinants of health, such as food security, housing, employment, social support, and gender affirmation, and not inherently related to sex work. Substantial or severe substance use problems may also impact engagement in PrEP care. While TW predominantly prioritized gender affirmation over PrEP services, the bidirectional benefits of gender affirmation and PrEP were noted. Colocation of services that address substance use, medical gender affirmation services, and linkage to other social services along with PrEP may facilitate retention in PrEP care, which future research should address. Additionally, research is needed to understand how structural factors—such as policies related to social determinants of health, sex work, substance use, and gender affirmation—impact transgender women’s access to PrEP and HIV health equity.

## Figures and Tables

**Table 1 ijerph-21-00133-t001:** Characteristics of persons assigned male at birth who identified as transgender women or nonbinary and initiated PrEP as part of the iMPrEPT study (CCTG 603).

					24 Weeks							
	Total (*N* = 170)				Retained (*N* = 114)				Lost to Follow-Up (*N* = 56)			
Characteristic	*n*	%	Mean	SD	*n*	%	Mean	SD	*n*	%	Mean	SD
Age in years	-	-	33.1	10.2	-	-	34.0	0.7	-	-	31.2	8.8
Race/ethnicity												
White	35	20.8	-	-	25	22.3	-	-	10	17.9	-	-
Latinx/Hispanic	60	35.7	-	-	35	31.3	-	-	25	44.6	-	-
Black/African American	24	14.3	-	-	17	15.2	-	-	7	12.5	-	-
Asian/Native Hawaiian/Pacific Islander	20	11.9	-	-	14	12.5	-	-	6	10.7	-	-
Other	29	17.3	-	-	21	18.8	-	-	8	14.3	-	-
Sexual orientation												
Heterosexual	85	50.0	-	-	49	43.0	-	-	36	64.3	-	-
Gay	9	5.3	-	-	6	5.3	-	-	3	5.4	-	-
Lesbian	7	4.1	-	-	6	5.3	-	-	1	1.8	-	-
Queer	10	5.9	-	-	8	7.0	-	-	2	3.6	-	-
Bisexual	19	11.2	-	-	16	14.0	-	-	3	5.4	-	-
Pansexual	26	15.3	-	-	19	16.7	-	-	7	12.5	-	-
Other	14	8.3	-	-	10	8.8	-	-	4	7.1	-	-
Relationship status												
Single	111	65.3	-	-	72	63.2	-	-	39	69.6	-	-
In a monogamous relationship	21	12.4	-	-	11	9.7	-	-	10	17.9	-	-
In an open/polyamorous relationship	23	13.5	-	-	20	17.5	-	-	3	5.4	-	-
Married	7	4.1	-	-	4	3.5	-	-	3	5.4	-	-
Separated, divorced, or widowed	4	2.4	-	-	3	2.6	-	-	1	1.8	-	-
Other	4	2.4	-	-	4	3.5	-	-	0	0.0	-	-
Highest level of education												
Less than high school	11	6.6	-	-	6	5.4	-	-	5	9.3	-	-
High school diploma or GED	46	27.7	-	-	26	23.2	-	-	20	37.0	-	-
Some college or technical training	83	50.0	-	-	59	52.7	-	-	24	44.4	-	-
Bachelor’s degree	16	9.6	-	-	13	11.6	-	-	3	5.6	-	-
Some post-graduate training	1	0.6	-	-	1	0.9	-	-	0	0.0	-	-
Graduate degree	9	5.4	-	-	7	6.3	-	-	2	3.7	-	-
Employment status												
Full-time employment	36	21.7	-	-	31	27.7	-	-	5	9.3	-	-
Part-time employment	30	18.1	-	-	24	21.4	-	-	6	11.1	-	-
Unemployed	81	48.8	-	-	45	40.2	-	-	36	66.7	-	-
Retired	2	1.2	-	-	2	1.8	-	-	0	0.0	-	-
Student	8	4.8	-	-	4	3.6	-	-	4	7.4	-	-
Unable to work (i.e., due to disability)	9	5.4	-	-	6	5.4	-	-	3	5.6	-	-
Engages in sex work as full or part-time employment	17	10.8	-	-	8	7.4	-	-	9	18.4	-	-
Health Insurance	116	68.2	-	-	83	72.8	-	-	33	58.9	-	-
Substantial or severe drug use problem (past 12 months)	16	10.9	-	-	8	7.8	-	-	8	17.8	-	-
Hormone therapy use	111	66.1	-	-	80	70.8	-	-	31	56.4	-	-

PrEP = pre-exposure prophylaxis; SD = standard deviation. Numbers may not sum to column totals due to missing data; Percentages may not sum to 100 due to rounding.

**Table 2 ijerph-21-00133-t002:** Characteristics of in-depth interview participants (*n* = 15).

Gender	Female/woman, *n* (%)	5 (33.3)
	Trans female/trans woman, *n* (%)	7 (46.7)
	Transfeminine, male to female, transgender, or transsexual woman, *n* (%)	4 (26.7)
Race and ethnicity	Black or African American, *n* (%)	3 (20.0)
	Latina/x, *n* (%)	6 (40.0)
	White, *n* (%)	9 (60.0)
	Spanish or multiracial, *n* (%)	2 (13.3)
Currently on PrEP, *n* (%)		10 (66.7)
Sex work experienced, *n* (%)		9 (60.0)
Age, mean (standard deviation)		35.8 (9.7)

Totals for gender and race/ethnicity sum to greater than 15 because participants were allowed to select more than one gender and race/ethnicity.

## Data Availability

The data are not publicly available to protect the confidentiality of the participants.

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
