# Peer review of "Determinants of HIV Pre-Exposure Prophylaxis (PrEP) Retention among Transgender Women: A Sequential, Explanatory Mixed Methods Study"

_ijerph, 2024, doi:10.3390/ijerph21020133_

Round 1
Reviewer 1 Report
Comments and Suggestions for Authors
Authors have studied the determinants of PrEP retention among transgender women living in Southern California.Though such studies are vital in order to restrict HIV spraed, I have the following comments/suggestions.
As authors have mentioned, participants from Phase 1 were not included in Phase 2. This for sure makes it difficult to correlate the findings of Phase I with Phase II. It would be nice if authors would come with the reason for doing so.
Novelty seems missing. The authors need to explain how and why their study and findings would add new information.
In the introduction section authors should include the background information on GATH in the context of PrEP.
The list of questions that was asked by the participants and their replies can be summarized in the form of a table. This would make the manuscript easy to understand.
The COVID and the lockdown might have been a factor affecting the study. The authors need to elaborate on that.
Author Response
Comment 1: Authors have studied the determinants of PrEP retention among transgender women living in Southern California. Though such studies are vital in order to restrict HIV spread, I have the following comments/suggestions.
Response: We thank you for highlighting the importance of our study.
Comment 2: As authors have mentioned, participants from Phase 1 were not included in Phase 2. This for sure makes it difficult to correlate the findings of Phase I with Phase II. It would be nice if authors would come with the reason for doing so.
Response: We have elaborated on our sampling design. This section of our methods now reads, “Participants in Phase-II were not recruited from Phase-I participants thus it is unlikely Phase-II participants were in Phase-I of the study. Although participants in Phase-II were likely not in Phase-I, they were recruited from the same target population. This is referred to as a parallel, rather than identical or nested, sampling design, which is appropriate for sequential mixed-methods studies.35 We chose to do so for feasibility.” (Lines 128-133)
Comment 3: Novelty seems missing. The authors need to explain how and why their study and findings would add new information.
Response: To enhance the clarity of the novelty of our findings, we have added to our discussion section. These sections are as follows:
“Previous research has found that structural factors, such as HIV stigma, history of incarceration, access to gender affirming care, health insurance, housing instability, and systemic racism are barriers to PrEP use among TW engaged in sex work.4,12,32 Our findings extend previous research by suggesting that lack of access to these social determinants of health are drivers of PrEP disengagement among TW engaged in survival sex work, however less so for those engaged in non-survival sex work.” (Lines 395-401)
“Not having to worry about HIV helped TW focus on their gender affirmation-related goals and similarly, not having to worry about gender affirmation-related goals could allow TW to focus on engaging in PrEP care. Thus, our study identified potential impacts of gender affirmation on PrEP and vice versa, suggesting a bidirectional relationship.” (Lines 424-428)
Comment 4: In the introduction section authors should include the background information on GATH in the context of PrEP.
Response: We have added to our background section on GAHT and PrEP. This section now reads, “Medical gender affirmation, such as the use of gender affirming hormone therapies (GAHT) and surgical procedures, has been associated with PrEP awareness and PrEP use.22-24 Qualitative research has suggested that lack of access to gender affirmation is an important determinant of PrEP care, and that co-location of services may help facilitate PrEP engagement.27,28 Further, concerns about drug-drug interactions (DDIs) between GAHT and PrEP have been noted by TW.29 Yet, research suggests that PrEP does not have an impact on GAHT, and although GAHT may reduce PrEP drug concentrations, this effect is small in magnitude and drug levels remained within the therapeutic range. However, more work is needed to understand how and why gender affirmation is related to PrEP use among TW.30” (Lines 79-88)
Comment 5: The list of questions that was asked by the participants and their replies can be summarized in the form of a table. This would make the manuscript easy to understand.
Response: While a complete list of the in-depth interview questions and a summary of participants’ replies may be of interest so some readers, we believe it would be more beneficial and parsimonious to provide example questions to our methods section, which now reads, “The interview guide included open ended questions, for example, “What have been your experiences with transactional sex?”; “What, if any, challenges have you faced with taking PrEP?”; and “If it were easier to access gender affirmation services, how if at all would this help you in terms of PrEP?”” (Lines 151-155)
Comment 6: The COVID and the lockdown might have been a factor affecting the study. The authors need to elaborate on that.
Response: We agree that COVID may impact our study findings, which is why we included it as a limitation. However, it is difficult to interpret how COVID may impact our results given it was not a point of inquiry in our study. We have expanded our limitations section around this topic which now reads, “Phase-II data were collected at the beginning of the COVID-19 pandemic, which may have impacted participants’ current experiences of sex work, substance use, and gender affirmation on PrEP engagement. However, few participants in Phase-II discussed the COVID-19 lockdown and its impacts on PrEP, limiting our ability to interpret how it may impact our study findings. Given our findings’ alignment to research conducted before COVID-19, this may suggest our results are robust to the time following the COVID-19 pandemic.” (Lines 454-460)
Reviewer 2 Report
Comments and Suggestions for Authors
Abstract (Lines 22-35)
- Clarity and Context: The abstract succinctly presents the study's methodology and findings. However, it starts directly with the method without a brief contextual background. For instance, the statement "In Phase-I, more participants not retained at 24 weeks..." (line 26) could be prefaced with a sentence summarizing why PrEP retention is a critical issue for transgender women.
- Detailed Suggestions: Include a brief introductory sentence in the abstract that sets the context for the study, like the high HIV prevalence among transgender women and the role of PrEP.
Introduction (Lines 38-80)
- Transition and Flow: While the introduction provides relevant statistics (e.g., "prevalence of HIV among transgender women (TW) to be 42%" on line 39), the transition to PrEP uptake is abrupt. A smoother link, perhaps through a sentence that connects HIV prevalence to the importance of PrEP uptake, would enhance readability.
- Citing Recent Data: References such as "Health inequities among TW can in part be attributed to cisgenderism..." (line 41) are well-placed, but the inclusion of more recent studies or data would reinforce the argument's currency.
Materials and Methods (Lines 81-146)
- Sampling Methods Detail: The section lacks detailed information about how participants were chosen for the study. For example, line 86 mentions "170 TW... enrolled in iMPrEPT", but how these individuals were selected or their characteristics need more detail for assessing bias and generalizability.
- Adherence to STROBE guideline : To better align with STROBE guidelines, this section could benefit from a more in-depth explanation of the study's design, particularly in the initial description of the mixed-methods approach (line 83).
Results (Lines 151-300)
- Linking Quantitative and Qualitative Findings: While both phases' results are clearly presented, the integration between the two could be more explicit. For instance, the transition from quantitative findings (e.g., "18% versus 8% reported substantial/severe drug use" on line 27) to qualitative insights (e.g., "participants reported drug use as a barrier to PrEP" on line 28) could be more seamless.
- Enhancing Scientific Rigor: The manuscript could detail the specific statistical methods used in the quantitative analysis to bolster scientific rigor.
Discussion (Lines 353-403)
- Structured Approach: The discussion effectively interprets findings but can be better organized. For example, discussions about "substantial or severe substance use" (line 373) could be grouped under a specific subheading for clarity.
- Current Literature: Ensure the discussion references the most current research, especially when discussing evolving areas like PrEP uptake strategies or transgender healthcare.
Limitations (Lines 404-413)
- Expanding Limitations Discussion: The limitations section briefly mentions potential generalizability issues but could be expanded. For instance, the statement "Our findings may not generalize beyond our study cohort" (line 405) can be detailed with specifics on cohort demographics or study design limitations.
Conclusions (Lines 414-420)
- Linking Findings to Implications: The conclusion succinctly summarizes key findings but lacks direct links to practical implications or future research directions. A more explicit statement connecting findings to potential policy changes or clinical practice improvements would be beneficial.
General Comments
- Grammar and Syntax: Minor grammatical errors throughout the manuscript, such as awkward phrasings or punctuation issues, should be corrected for improved readability.
- Consistency in Argument: While the manuscript maintains a consistent argument, linking the sections more explicitly would enhance the narrative flow. For example, the connection between the "Introduction" and "Materials and Methods" could be more fluid.
Comments on the Quality of English Language
Minor editing of English Language required. The manuscript is generally well-written, but a careful proofreading to correct minor grammatical errors would improve readability.
Reviewer 3 Report
Comments and Suggestions for Authors
Summary:
This study used a two-phase mixed methods approach to investigate factors affecting PrEP retention among transgender women (TW). Phase I analyzed data from 170 TW on PrEP, finding lower retention rates among those engaged in sex work and with substantial drug use. Phase II involved interviews with TW, revealing that drug use and the nature of sex work (especially survival sex work) were significant barriers to PrEP adherence. Additionally, fewer participants not retained in the study were using gender-affirming hormone therapy. The findings suggest that TW involved in survival sex work face unique challenges in maintaining PrEP adherence and may need additional support. While the study presents robust findings, there are certain areas that require further attention to enhance its comprehensiveness.
Minor:
1. In the introduction, it would be beneficial to provide a more detailed exposition on HIV PrEP. This should include an in-depth introduction to PrEP's role and efficacy in HIV prevention. Additionally, it is imperative to elucidate the significance of comprehending PrEP retention, particularly among TW, highlighting how this understanding can contribute to more effective HIV prevention strategies.
2. Please ensure consistency in the usage of terms throughout the paper. For example, on line 106, it is written as “For Phase-I”, whereas on line 126, it appears as “For Phase I”.
3. Please ensure uniformity in capitalization. For instance, in Table 2, "Trans female/trans woman" is capitalized, while "transfeminine" is not.
4. Line 190, please provide the full term for the abbreviation "STIs". It should be expanded to "Sexually Transmitted Infections" for clarity and completeness.
5. Line 267-268, “(006, 28, on PrEP, engaged in sex work)”, is unclear. Could you please clarify the meaning of “006” and “28” in this context?
Comments on the Quality of English Language
Minor editing of English language required
Author Response
Please see the atttachment

Round 2
Reviewer 1 Report
Comments and Suggestions for Authors
The authors have addressed the reviewers comments. Although a minor editing in the language would help improve the clarity of the article.
For example:
"iMPrEPT was a 103 48-week randomized controlled trial evaluating brief motivational interviewing added to 104 a text message-based adherence intervention among 255 transgender individuals (those 105 assigned female sex at birth were excluded from the current analysis) in San Diego and 106 Los Angeles provided oral PrEP with emtricitabine/tenofovir disoproxil fumarate 107 (FTC/TDF) through study starting at baseline from June 2017 to September 2020.
Comments on the Quality of English Language
Overall the language can be edited in the manuscript for better quality.
Author Response
Reviewer 1: The authors have addressed the reviewers comments. Although a minor editing in the language would help improve the clarity of the article.
For example:
"iMPrEPT was a 48-week randomized controlled trial evaluating brief motivational interviewing added to a text message-based adherence intervention among 255 transgender individuals (those assigned female sex at birth were excluded from the current analysis) in San Diego and Los Angeles provided oral PrEP with emtricitabine/tenofovir disoproxil fumarate (FTC/TDF) through study starting at baseline from June 2017 to September 2020.”
Response: Thank you for your critical feedback on our manuscript. We have made efforts to edit the manuscript to improve clarity. For example:
“iMPrEPT was a 48-week randomized controlled trial evaluating brief motivational interviewing along with a text message-based adherence intervention among 255 transgender individuals in San Diego and Los Angeles whom were provided oral PrEP with emtricitabine/tenofovir disoproxil fumarate (FTC/TDF) throughout the study starting at baseline from June 2017 to September 2020.32 Participants assigned female at birth were excluded from the current analysis.” (Lines 103-110)
Reviewer 2 Report
Comments and Suggestions for Authors
No further comments as authors have addressed the issues raised earlier
Comments on the Quality of English Language
Minor English corrections
Author Response
Reviewer 2: No further comments as authors have addressed the issues raised earlier
Response: We thank you for your critical feedback.